# "It's Like Having an Uncontrolled Situation": Using Body Maps to Understand the Embodied Experiences of People with Hidradenitis Suppurativa, a Chronic Dermatological Condition

**Natalie Ingraham** [1,*] **, Kelly Duong** [2] **and Lena R. Hann** [3]

[1] Department of Sociology and Anthropology, Farmingdale State College, Farmingdale, NY 11735, USA
[2] Department of Sociology, College of Social Sciences, Honolulu, HI 96822, USA; kduong3@hawaii.edu
[3] Public Health Program, Augustana College, Rock Island, IL 61201, USA; lenahann@augustana.edu
[*] Correspondence: natalie.ingraham@farmingdale.edu

**Abstract:** Hidradenitis suppurativa (HS) is a chronic, inflammatory, and often debilitating skin condition that includes painful "flares" in the groin, genital, and underarms. (1) Background: Patients with HS have the highest reported mental health comorbidities among dermatological conditions. Qualitative social science research about HS is limited, so this study aimed to understand the lived experiences of people with HS through body mapping. Body mapping is a participatory research process where participants illustrate a drawing of their body with images, symbols, and words that represent their embodied experience. (2) Methods: This study recruited 30 participants from a previous survey about HS experiences. Participants selected from pre-made body silhouettes based on their body shape, illustrated a body map about their HS experience, then shared their body map during in-depth interviews. Interviews and body maps were analyzed with the same codebook created with inductive and deductive codes. (3) Results: The body map drawings yielded rich visual data and the mapping process helped participants express their HS experiences in unique ways that cannot always be captured with textual data alone. (4) Conclusions: This study adds to the limited social science literature about HS and introduces body mapping as a relevant qualitative method for exploring chronic dermatological conditions.

**Keywords:** body mapping; dermatology; chronic illness; qualitative methods; visual data





## 1. Introduction

*Then I let go of the shame, knowing I was not to blame. It did, it did have a name. It was, it is, HS that caused such great pain.—Denise (cisgender woman, White/Native American, USA)*

This study uses body mapping as a qualitative and interdisciplinary visual social science method to understand the lived experiences of people with hidradenitis suppurativa (HS). HS is a chronic, inflammatory, debilitating skin condition that impacts an estimated 1–4% of the population. Often beginning in puberty, HS is characterized by painful, deep, and inflamed lesions or "flares" mainly in the groin, perineum, genital, and underarm areas, though other areas of the body can be affected such as the head, neck, or ears (Ballard and Shuman 2024; Margesson and Danby 2014). There are no current cures for HS and treatment efficacy varies widely (Alikhan et al. 2009; Macklis et al. 2022; van Straalen et al. 2020). HS patients also have the highest reported mental health comorbidities among dermatological conditions and report high levels of emotional distress with increasing severity of the condition (Matusiak 2020). Using the body mapping method to study the lived experience of HS patients builds on previous work in the use of drawings, illustrations, and other visual methods to study illness and health (Guillemin 2004; Phillips et al. 2015; Guillemin and Drew 2010) and body mapping studies from fields like psychology,

art therapy, public health, and geography (Jokela-Pansini 2021; Boydell et al. 2020; De Jager et al. 2016). Because HS is a particularly visual and physical condition—it often presents with redness, inflammation, intense scarring, and pain—we wanted to give participants a visual outlet for expressing their embodied experience of living with HS. In the following sections, we explain key concepts about hidradenitis suppurativa, outline previous research on drawing and body mapping as visual research methods, and describe our methodology for data collection and analysis. We describe the empirical results of the body map analysis and its implications for people with HS and other social scientists studying chronic dermatological conditions. Like in previous studies (Guillemin 2004; Guillemin and Drew 2010; Coetzee et al. 2019), we also describe the benefits and limitations of using body mapping methodology as part of a mixed methods research study on chronic illness.

*1.1. Background*

1.1.1. Hidradenitis Suppurativa

Hidradenitis suppurativa (HS) is a painful, chronic, autoinflammatory dermatological disease with no known cause or cure, with treatments ranging from lifestyle recommendations (like losing weight or quitting smoking) to pharmaceutical prescriptions (including biologics) to surgery (Ballard and Shuman 2024; Alikhan et al. 2009; Macklis et al. 2022). Many people who have HS do not receive an accurate diagnosis until over 10 years after symptoms first appear, which can lead to greater physical and emotional suffering as the disease progresses (Lee et al. 2017). This is a prevailing issue in the medical field because HS is still relatively unknown among patient populations, meaning the first symptoms are often misidentified as ingrown hairs, acne, a hygiene issue, or even herpes, which both delays proper diagnosis and treatment and can exacerbate stigma (Editorial Team HS Disease.com 2020). Even more, general practitioners only correctly diagnose HS 20.4% of the time while family physicians self-report low confidence in HS diagnosis and treatment (Collier et al. 2020; Esme et al. 2021). Dermatologists are best equipped to diagnose and treat HS, but barriers to training, medical referrals, and access to specialty care can prevent patients from seeing a dermatologist knowledgeable about HS in a timely manner (Schukow et al. 2023).

HS symptoms, also known as flares, can last weeks, months, or years, and can cause physical discomfort and pain, foul smell from the wound site, infection, emotional distress, and social isolation (Alikhan et al. 2009; Sabat et al. 2020). At its worst, HS can be completely debilitating, resulting in permanent physical disability that can impact employment, family engagement, and other quality of life factors (Patel et al. 2017; Tzellos et al. 2019). One common way HS is classified is by Hurley stages (I, II, or III), an often-unreliable scale that measures the severity and progression of the disease throughout the body (van der Zee and Jemec 2015). People with HS at all levels, but especially at Hurley stages II and III, report that the disease impacts their quality of life and that the stigma of HS impacts interactions with partners and providers, contributing to higher rates of anxiety, embarrassment, and depressed mood (Kimball et al. 2024; Ingraham et al. 2022). HS is more commonly diagnosed among cisgender women than cisgender men, with a higher prevalence among people of color compared to their White counterparts (Garg et al. 2017). As such, HS carries a greater disease burden for marginalized populations (Kilgour et al. 2021).

1.1.2. COVID-19 and HS

The height of the COVID-19 pandemic presented unique challenges for people with HS. Changes in clinical operations, availability of surgical procedures, and access to medical management disrupted patient access to care, as well as treatments to reduce HS flares, pain, and discomfort. Since HS is an autoinflammatory condition, HS patients were not sure if SARS-CoV-2, and later the vaccine to prevent it, presented additional health risks and thus several reported socially isolating themselves to prevent exposure to the virus (Dovalovsky et al. 2023). Dermatologists implemented telemedicine appointments during this time,

including privacy-protecting online platforms to share photos of flares, to meet patient needs when in-person appointments were not possible or not safe (Dovalovsky et al. 2023). However, due to HS's impact on social and emotional well-being, the pandemic and related lockdowns exacerbated patients' physical and mental health concerns, and many turned to social media groups for guidance and support (Yesantharao et al. 2023). Our study was designed in the early months of the pandemic and acknowledged the multifaceted barriers HS patients faced, not only regarding their health and social connectedness but pandemic-related issues like what has now been studied as "Zoom fatigue" (Nesher Shoshan and Wehrt 2022). As such, we wanted to explore HS in a new way that creatively engaged participants and added to sociological and public health understandings of the disease's impact on embodied, lived experience.

### 1.1.3. Theoretical Frameworks

The primary purpose of this study was to explore the lived experience of people with HS and how it shapes their lives and relationships. We draw from a rich literature in the sociology of health and illness that examines illness experiences and narratives (Charmaz 1991; Bury 1982). These interpretive approaches, guided by symbolic interactionism, explore meaning making for people with chronic illness and the ways that a diagnosis of chronic illness creates "biographical disruptions" that changes both social relationships and daily activities (Bury 1982). Charmaz's (1991) work demonstrated how chronic illness shifts people's sense of time and how their sense of self is also reconstructed, which is particularly applicable for a chronic, progressive condition like HS. Charmaz (1983) also contributes new understandings of loss of self as a type of suffering for chronically ill people, in that the social isolation, restricted lives, and burdening of others also generate suffering beyond physical discomfort. Like Jeske et al. (2023), we also wanted to emphasize the importance of studying illness and diagnosis as a temporal experience that can impact interpersonal relationships by asking participants about the progression of their HS over their lifetime rather than only considering their current experience with the condition (Le Henaff and Heas 2023; Richardson et al. 2007). Previous work applying Bury's concept of biographical disruptions to dermatological conditions (Jobling 1988; Al-Muhandis 2024) also shaped our understanding of how HS might present a unique type of biographical disruption as a dermatological chronic illness that could be illustrated using body maps.

### 1.1.4. Visual Methodology in Qualitative Research
Drawings

Many scholars have called for the expansion of person-centered qualitative research approaches, including the use of visual methodologies that examine participant experiences. Drawings have been used across disciplines to investigate both the processes and outcomes of meaning making (Guillemin 2004; Brailas 2020). Health and illness research benefits from using drawing in combination with other qualitative methods to expand "our interpretations as researchers of the many, diverse ways in which illness can be understood and experienced" (Guillemin 2004, p. 286). Using drawings benefits participants by enabling deeper engagement, increasing their sense of visibility within the topic being studied, and allowing more nuance to capture experiences that are difficult to describe verbally (Guillemin and Drew 2010). Similarly, drawing as a method helps researchers extend their methodological repertoire, invites more types of data that can be produced and analyzed, and creatively engages in new ways to produce knowledge (Guillemin and Drew 2010; Brailas 2020).

While our extensive literature review did not yield any published studies about using drawing methods with HS participants, researchers have used drawing to understand patient experiences with chronic pain. Phillips et al. (2015) found that patients who were encouraged to draw representations of their chronic pain produced vivid and emotionally charged drawings that facilitated a "visible and shareable language for pain" (p. 410). As with all qualitative research methods, drawing should be approached with keen ethical

considerations about its relevance to the population studied, what other emotions may come up during drawing, and what will happen to the drawings that are produced. Transparency, trust-building, and practicing the method prior to data collection are all important for incorporating drawing into qualitative mixed methods research (Brailas 2020; Guillemin and Drew 2010).

1.1.5. Body Maps

Body mapping is another visual qualitative method that focuses on both the process and outcome of the research approach. It is a relatively new type of participatory qualitative research that yields rich data about participants' embodied experiences that may be difficult to understand through other methods like interviews or ethnography. Body mapping has been used extensively in art therapy and as an advocacy tool, as well as for research in the health sciences, psychology, and geography, among other disciplines. For an in-depth history and systematic review of body mapping in academic literature, see De Jager et al. (2016).

In the social sciences, body mapping facilitates visual storytelling through participants' lenses and how they see and experience themselves in the world (Coetzee et al. 2019; Gastaldo et al. 2018). Traditionally, the body mapping methodology includes three components: (1) a life-sized body map, where the participant traces their body on a large sheet of paper and designs/decorates it according to prompts from the session leaders; (2) a *testimonio* or short story that the participant creates to narrate the meaning and importance of their body map; and (3) a key that helps explain symbols, colors, and other dimensions of the art on the body map. Most importantly, completed body maps must include an explanation or narrative by the person who created it, thus contextualizing the map in the participants' lived experience (Coetzee et al. 2019).

Body mapping has been used globally to understand embodied experiences of health and illness, including in studies on chronic conditions and pain. These include young people with chronic and somatic pain in the Netherlands (Van Schelven et al. 2023), impacts of long COVID on people from the U.S. (Santarossa et al. 2023), reflections on birthing in India (Mayra et al. 2022), experiences of dialysis treatment in Canada (Ludlow 2014), and pain and injury among professional dancers in the United Kingdom (Tarr and Thomas 2011). HS is a chronic health condition with painful physical symptoms that can impact emotional and social outcomes. Body mapping, as one method in our multi-method study, is appropriate for understanding HS because it presents interactional and artistic opportunities, allowing for the expression of nuanced embodied experiences that are not always easily explained with words (Coetzee et al. 2019).

Traditional body mapping is a time- and resource-intensive methodology that engages researchers and participants through in-depth collaboration and meaning making. Body maps are usually produced during multi-session group workshops or on an individual basis in connection with focus groups or interviews (Coetzee et al. 2019). The COVID-19 pandemic impacted in-person research methods and several body mapping studies had to adjust their approaches accordingly. These included mailing materials to participants and hosting the workshops online (Vaughan et al. 2022; Santarossa et al. 2023) and designing a chatbot that helped participants design a digital body map (Van Schelven et al. 2023). To our knowledge, our project is the first to design a body mapping study during the height of the pandemic and adapt the method to a population already living within pandemic-era restrictions. Also, to our knowledge, this is the first study to use body mapping with participants with HS. Deeper reflection on how and why we adapted our body map methodology for an HS-affected population during the COVID-19 pandemic will be analyzed in another paper.

## 2. Materials and Methods

The HS Body Mapping Study involved three components: a quantitative survey, a body map illustration, and an in-depth interview. Participants for the survey were recruited through a community-based convenience sample of HS online support groups, social

media, and via a website and Instagram account created for the study. The co-principal investigators strategically disclosed (DeVault and Gross 2012) their status as people living with HS on the website and Instagram page to build rapport with potential participants who may have been skeptical of non-physicians requesting research participation. A total of 401 individuals began the survey; 263 completed at least 80% of the survey and were included in the final analyses. The survey analysis and results can be found in our first study publication (Ingraham et al. 2022).

Survey participants could opt-in to be contacted about additional data collection at the end of the survey. Fifty participants were drawn from this survey sample. We utilized purposive sampling to prioritize interviews with people of color and men, since the survey sample was overwhelmingly White, cisgender women. Qualitative interviews were completed from 2021 to 2022 and included instructions to complete a digital or paper body map before the interview. Participants chose a body map silhouette from 10 options of varying shapes and sizes. Each body shape had a front and back silhouette. The bodies were not specifically gendered, as none of them showed defined genitals, but half of the silhouettes had defined breasts. Participants were instructed to choose a shape that most closely resembled their physical body and use the front and back designs to create their body maps on their own before the interview. The interview guide covered four key areas based on our theoretical frameworks and literature review specific to HS and body mapping: HS history, provider interactions, romantic partner interactions, and body mapping process and explanation. Participants were also asked to talk the interviewer through their body mapping process and the imagery represented in the body map. The study was approved by the CSU East Bay IRB.

### 2.1. Qualitative Analysis

This analysis draws from participants' body mapping discussions from the in-depth interviews and the research team's analysis of the body maps themselves. Both were analyzed using Dedoose, an online mixed methods software. Participants were tagged with demographics in Dedoose called descriptors for country or region (USA, Canada, and Europe), gender (man, woman, and nonbinary), and race (Black, White, Latinx/Hispanic, and Biracial/Multiracial), though some participants did not disclose their racial identity during the interviews. Pseudonyms are used for participants alongside their demographics for gender, race, and country of origin, when available.

### 2.2. Interview Analysis

In-depth interviews were transcribed verbatim using a combination of traditional manual transcription by research assistants and the cleaning of Zoom's automated transcriptions for accuracy. Interview field note summaries were completed for each interview that summarized key points of the interview based on sections of the interview guide as well as informal observations about the interview process such as tone of voice, body language during video interviews, and reflexivity components on the flow of the conversation or interview disclosure of personal stories related to HS. Issues with technology were also noted, especially where they impacted transcription. The interview guide sections and these field notes were the source of an initial set of deductive codes. Initial coding was completed by the first author and two undergraduate research assistants trained in qualitative methods. Coding combined deductive codes drawn from the literature and interview guide questions and inductive in vivo coding from participant data directly. Deductive codes focused on the history and diagnosis of HS as key aspects of illness narratives, impacts on daily life, and patient–provider and patient–partner relationships. Axial coding was used to distill key themes from open coding. Themes related to the body map are highlighted in this paper, as the larger interview themes are outside the scope of this article.

*2.3. Body Map Analysis*

Body maps were analyzed by the first and second authors using visual sociology and body map analysis techniques (Cross et al. 2006). These include analysis of colors, shapes, or objects drawn in the body map and consideration of the audience for the map. Dedoose allows for text and image analysis. Image analysis is conducted by selecting an area of the image then considering that segment of the image an excerpt, like a sentence in an interview transcript, which can then be coded. We used the same codebook as the interviews to start with and added new in vivo codes for unique phrases written on the body maps. We created new parent codes for body map colors and body map images. Additional codes were added to other areas of the codebook as well, such as additional mental health or feeling words included on body maps. We also reviewed each participant's transcript in relation to their body map using keyword searches for "body map" to aid in coding participants' explanations of their body map colors, images, or words used.

Thirty-one of our fifty interview participants completed body maps. Although there were no specific shared characteristics of those who did not complete the body maps, most preferred telephone to Zoom interviews and mentioned struggles with the technological aspect of the process. This is an important limitation to consider for this method, especially for lower technology access or lower literacy populations in general who may benefit from in-person data collection when using visual methods like this.

## 3. Results

*3.1. Participant Demographics*

The subset of interview participants who completed a body map (*n* = 31) consisted of cisgender women (90%) from the United States (87%). Body map drawing participants were majority White (47%), followed by Latinx/Hispanic (17%). See Table 1 for a detailed comparison between the larger sample of interview study participants and the sub-sample of body map participants.

**Table 1.** Participant demographics.

| Demographic | Interviews Count (%) n = 50 | Body Maps Count (%) n = 31 |
|---|---|---|
| Gender | | |
| Cisgender women | 46 (92%) | 28 (90%) |
| Cisgender men | 3 (6%) | 2 (7%) |
| Nonbinary | 1 (2%) | 1 (3%) |
| Race | | |
| White | 24 (44%) | 15 (48%) |
| Black | 7 (14%) | 3 (10%) |
| Latinx/Hispanic | 8 (16%) | 4 (17%) |
| Biracial/Multiracial | 4 (8%) | 3 (10%) |
| Middle Eastern/North African (MENA) | 1 (2%) | 1 (3%) |
| Missing data | 7 (14%) | 4 (13%) |
| Country of Origin | | |
| USA | 42 (84%) | 27 (87%) |
| North America | 2 (4%) | 1 (3%) |
| Europe | 2 (4%) | 2 (7%) |
| Central America | 2 (4%) | 1 (3%) |
| Missing data | 2 (4%) | – |

Participants utilized a combination of colors, images, and words to convey nuanced meanings and emotions related to their lived experiences with HS, while also incorporating medical depictions. Participants tended to create one of two types of body maps: (1) colorful

and detailed illustrations of one or more objects or colors to describe living with HS and its impacts on various parts of their bodies and lives, and (2) medical-style body maps that were simple drawings that only indicated areas of the body impacted by HS with dots, Xs, or circles. Below, we describe the most common colors, images, and words used in the more illustrative body maps and the use of body maps as medical drawings with body map examples for each theme.

### 3.2. Body Map Colors

The most common colors used in body maps were red/orange, black, and blue. Twenty participants who used red (67%) described this color as being closely tied to the pain, inflammation, and/or scarring caused by the HS flares. Participants also used red to represent anger at managing their HS or at the pain or scarring it caused. Red and orange were often associated with images such as flames and dripping blood. Participants who engaged in a medical-oriented approach towards body mapping utilized red to represent scarring, flares, and blood as symptoms of HS. Areas of the body that were frequently colored red were the groin, armpits, and under the breasts, where HS flares are common. Grace (cisgender woman, no race disclosed, USA) characterizes red as, "fire coming out of the armpits, it fricking hurts! There's a constant burning sensation under both my arms. All the time. And you know living with that is just excruciating! Then [there's] just the red marks all the way around [to] indicate overall pain."

As visible in Grace's body map (Figure 1), red was used in several places to indicate pain, heartbreak, and scarring. Like Grace, Ernesto (cisgender man, Latino/Hispanic, Mexico) also used red and some orange to highlight key areas of HS activity including his armpits and chest (Figure 2). Ernesto described his choice for this color as follows: "Orange, because it really feels like having something on fire under the under arms now, not that I'm really in pain or something like that, but it's like having an uncontrolled situation, you know, just happening there." For Ernesto, the warm, fire-like colors were less about pain and more about the lack of control for the HS "situation," his flares.

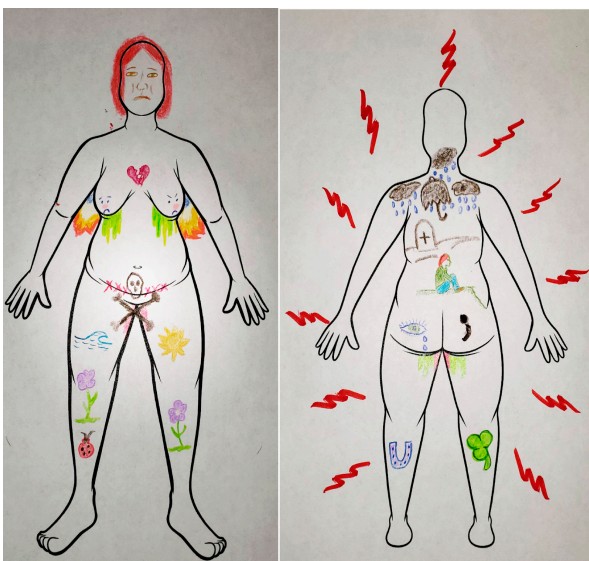

**Figure 1.** Grace's body map.

Twenty participants used black as an equally common color (67%) as red and orange. Because black is a base color that is often neutral, participants frequently chose black when using words to convey a diverse range of meanings. Black dots varied between scattered and concentrated but tended to cluster around areas impacted by HS, like the underarm. Black dots or Xs were sometimes used in "private" parts of the body, especially the groin, to represent vulnerability or shame associated with either romantic partners or providers touching those areas. Daria (cisgender woman, Iranian, USA) detailed why she correlated

black with the groin area (Figure 3), explaining, "I put barbed wire, especially around the groin because I've had the most traumatic experiences there." Black drawings around the groin were symbolic representations of private areas that are deemed "off-limits" for sexual intimacy due to the physical and emotional repercussions of having HS.

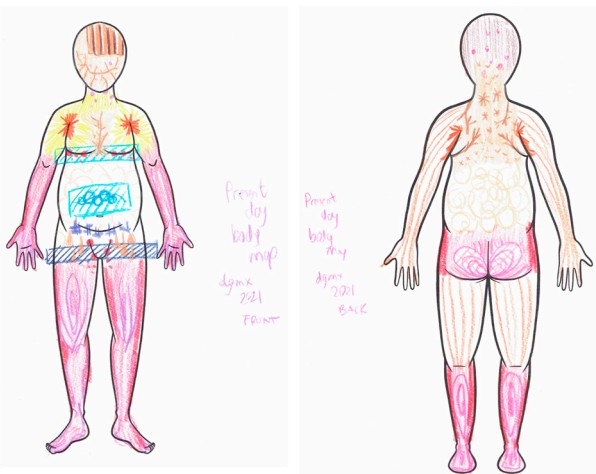

**Figure 2.** Ernesto's body map.

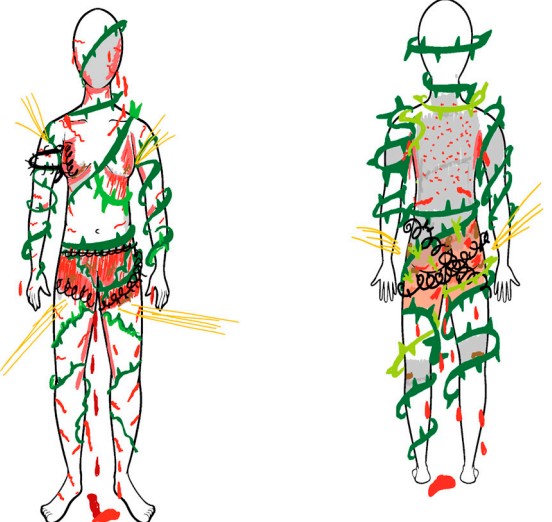

**Figure 3.** Daria's body map.

Black dots were used in several medical drawings for body maps to depict regions affected by HS, including prominent rashes and localized areas prone to flaring. Sarah (cisgender woman, White, USA) explained how she used dots in her body maps, stating, "I marked where I get them and then where the rashes tend to spread and I have some scarring on my stomach from like small ones that come through. . . The black is the previous scars in the (area of my) groin between my legs." Sarah provided a key for her body map (Figure 4) to describe where and how she chose the colors for each section impacted by HS.

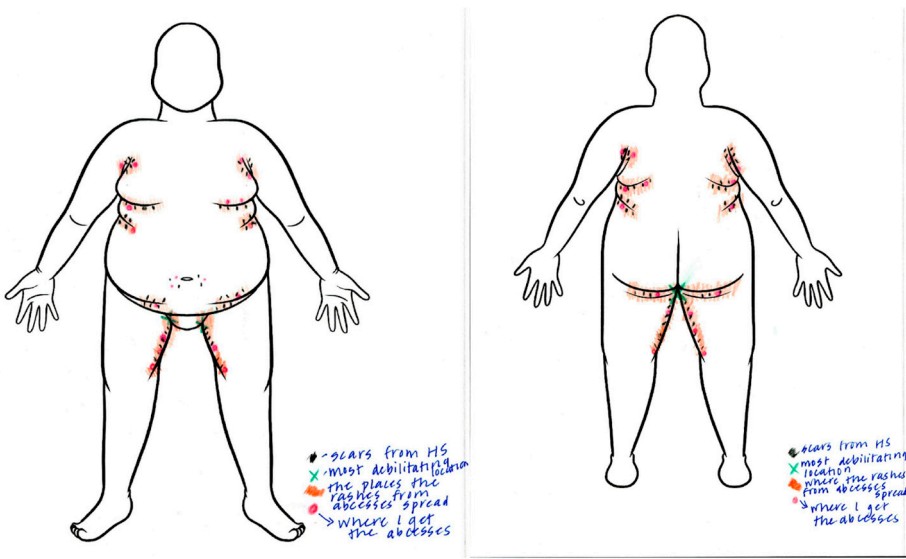

**Figure 4.** Sarah's body map.

Eimar (cisgender woman, White, Ireland) began her body map with red dots, but ended up covering her whole body in red then adding black dots to her HS affected areas later (Figure 5). She said that at one point she considered turning the whole body from red to black to represent feelings of shame. She shared, "you're constantly like, you have to cover this, you have to highlight this and you have to hide that." For her, the black dots represent areas of shame and the desire to hide her HS and her whole body. While Eimar made clear illustrations of her physical suffering, she also notes what Charmaz (1983) describes as the "loss of self" suffering that comes along with the isolation and shame of chronic illness that disrupts social interactions. Eimar felt shame about how others would see her HS and used a large, grey hand to visualize preventing others from interacting with her or her body.

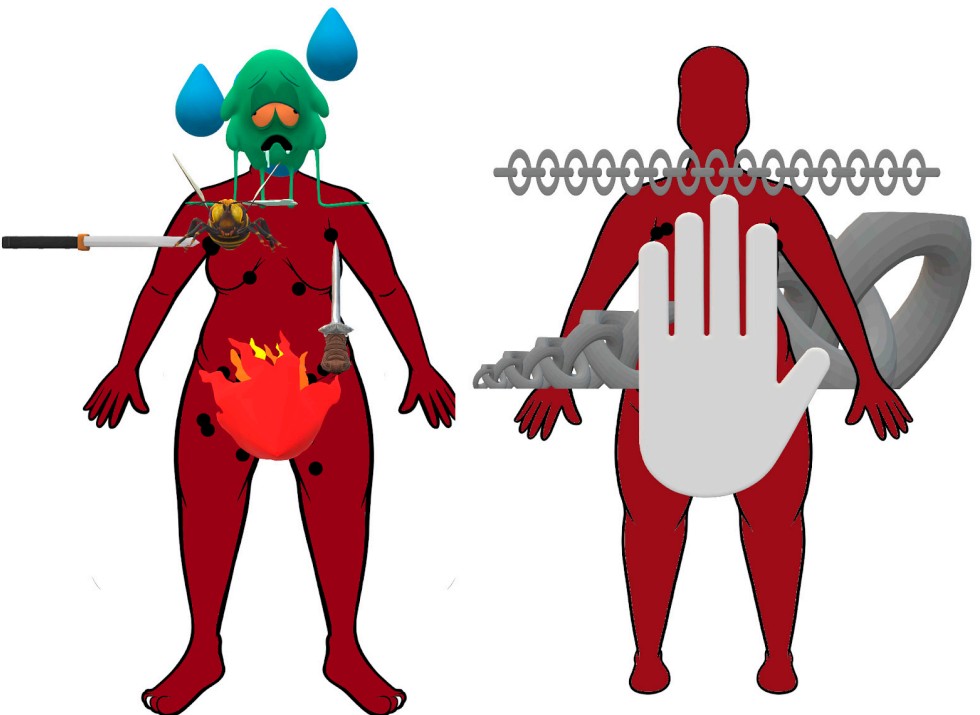

**Figure 5.** Eimar's body map.

Forty-three percent of participants utilized the color blue to express feelings of sadness, isolation, hopelessness, and shame. Blue was frequently applied around the face and head to depict tears and sweating. Denise (cisgender woman, White/Native American, USA) described her experience with the body map illustrations and why she drew images of tears using blue (Figure 6):

> *This was something that was an issue over the years. So "tears I've cried" because of shame until I learned what caused my pain. . . .And I did a lot of sweating and I've spent a lot of time crying. And so I put the tears, and the puddles on the floor. Because, you know, you're crying because there just wasn't awareness. When I was younger, I felt such shame and I felt like I had to hide things, and so I remember going through that.*

Here, Denise highlights the ways that HS has impacted her "over the years", indicating the lifetime impact of HS as a progressive condition. She describes the onset of HS at age 10, but the progression and worsening of the HS in her 20s and eventual diagnosis in her 30s led to more of Bury's concept of biographical disruption. She describes "reading everything" about HS after her diagnosis and links it directly to her other chronic inflammatory conditions that have resulted in disability. Denise also drew a purple ribbon, which is widely considered the awareness color for HS and is often used on HS websites or on social media posts by HS patients with the hashtag "HS warrior." For her, the ability to connect with other HS patients and identify as a warrior represents a shift in her sense of self—from one focused on shame before her diagnosis to a more active participant in her HS "fight."

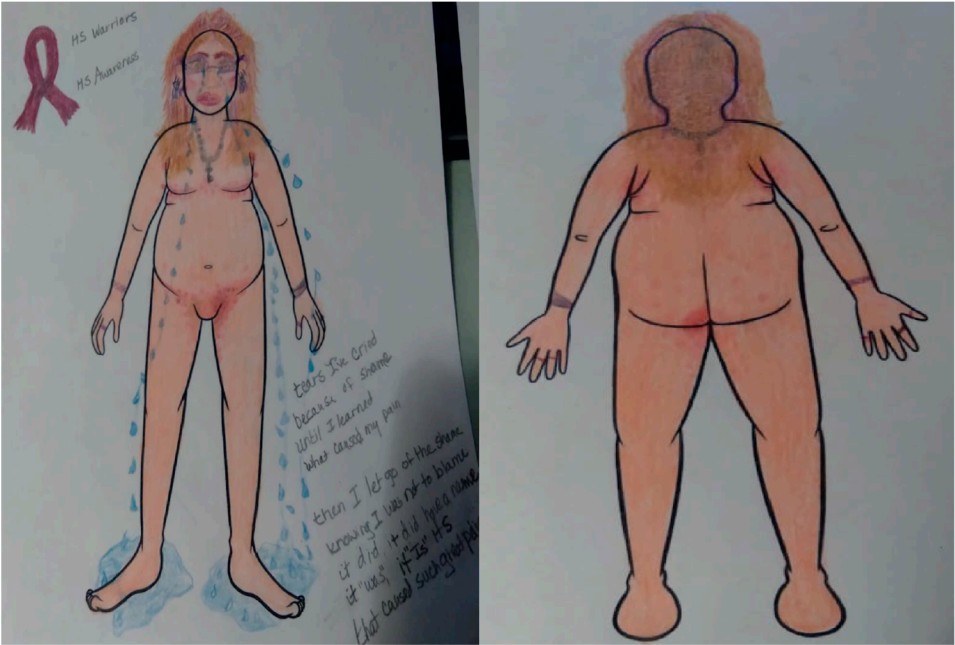

**Figure 6.** Denise's body map.

In contrast, blue was also used to represent feelings of healing and acceptance, perhaps attributed to its cooling effects. Eliza (cisgender woman, Latina/Hispanic, USA) described blue as, "sort of comforting. . . it's like, this part of my body I am fine with and I don't feel the urge to change." Eliza (Figure 7) only illustrated the front of her body figure by featuring her HS journey across time, which was not part of the instructions. She used separate color palettes for each time period that showed shifts in HS-impacted areas and the worsening of pain and scarring. For example, the lower abdominal area moved from a blue to orange color to represent new HS flares in her current body versus the one from several years earlier. Eliza's two drawings are a clear representation of the biographical disruption of HS to her life. Her earlier 2008 body map demonstrates how HS made her feel confused, "disgusting and filthy" in ways that left her with a "general sense of distaste"

for herself. She said completing the body maps felt like she was taken right back "to the first flare, like first issue" and how she was feeling. Interestingly, although her HS has worsened in some areas, her sense of self and relationship to her body has actually grown more positive over time and after she had her two children, as reflected in her 2021 map. Other participants reflected on their HS progression over time, but Eliza was the only person to illustrate it with multiple body maps and color changes.

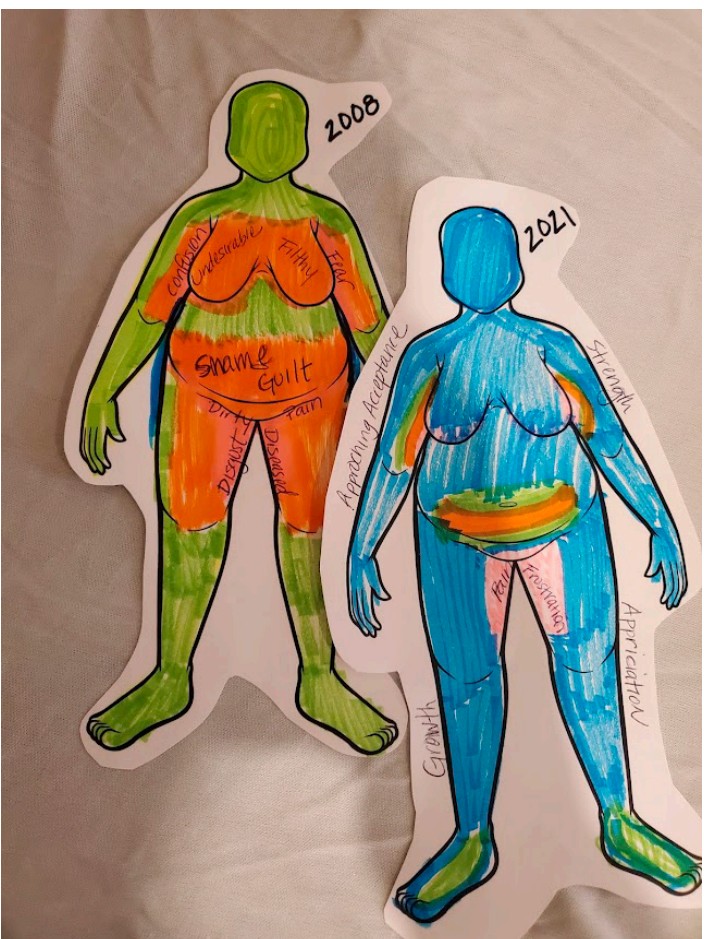

**Figure 7.** Eliza's body map.

Many participants used materials they had available to them at home; the research team did not mail body map materials like in other studies (Vaughan et al. 2022; Santarossa et al. 2023). As such, several participants used common household items and office supplies like highlighters, pencils, and pens, and thus their color palette to illustrate body maps was more limited. Others used more artistic materials like crayons and colored pencils, and several participants used features on their computer or phone to illustrate their body maps. Cooler colors like blue or purple represented happiness or neutral feelings about HS-free parts of their bodies. Warmer colors like pinks, oranges, and yellows were often used in HS-impacted areas along with red to represent scarring and pain. Pain, scarring, and emotional challenges were not only represented by colors, but also by specific images drawn by several participants.

*3.3. Body Mapping Images*

Body mapping images are closely connected to the colors used and most often represented types of pain and life limitations felt by participants due to HS. Pain imagery included flames and knives. For example, Eimar (Figure 5) drew several types of knives and swords to represent the different types and depths of her HS pain. Daria (Figure 3)

drew thorns around her whole body but noted that rather than representing a type of HS pain, it was more about protection. She said, "That's not necessarily where I have HS, but I just kind of feel like it takes a lot for you to touch me. So I don't like being touched. I'm not a hug person." Daria's distancing of herself from others physically and emotionally represents Charmaz's (1983) argument that chronic illness produces social isolation and discrediting of the self. For Daria, this means her HS has limited her romantic and daily life in terms of physical touch and emotional closeness.

The limitations of HS on daily living and larger life goals were illustrated with chains, a person behind jail cell bars, and cross-out symbols. Andrena (cisgender woman, Latina/Hispanic, USA; Figure 8) described how intensely HS limited her life by using cross-out symbols for how her travel, social, sexual, and professional experiences are impacted, a broken key to represent her inability to escape from HS, and a black heart to represent a "death" of feelings, including her HS-related depression. She said, "I feel like I am in a prison and someone locked [me] in. . . and br[oke] the key. . . Because, um, we don't know how to deal with this, how to control it."

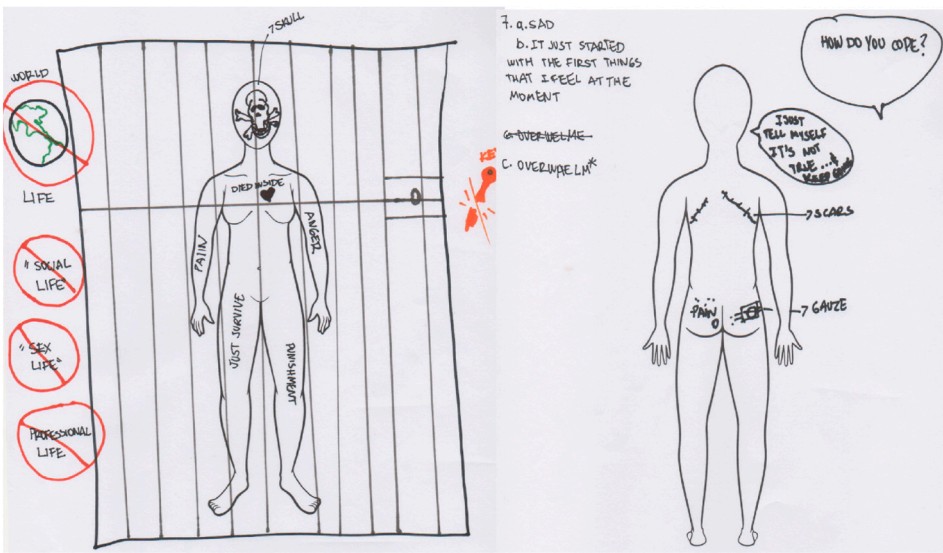

**Figure 8.** Andrena's body map.

These limitations or crossed-out images represent how HS impacts loss and grief over activities participants would like to do or life goals they would like to meet, such as having a more active social or sexual life. This portrays the ways that chronic illness disrupts not just daily habits like showering or eating but also the potential for relationships in the person's social network, especially romantic ones. Andrena also highlights the potential loss of capital resources resulting from chronic illness (Bury 1982) due to limitations on professional life when HS makes it harder or sometimes impossible to work.

Emotional pain was illustrated as crying tears, broken hearts, or black and gray storm clouds. Like Grace (Figure 1), Laura (cisgender woman, White, Italy) also used a broken heart to describe the challenges of dating with HS and the emotional pain and shame of the condition (Figure 9).

Many participants also included positive images, especially for parts of their bodies not impacted by HS. Ernesto (Figure 2) drew tree ring-like images on his legs that represented strength and beauty, as they were his girlfriend's favorite part of his body. Grace (Figure 1) drew flowers and ladybugs on her legs because of positive feelings toward that part of her body. She said, "I'm gonna be honest, it's the only part of my body that I actually really like and [my legs] are fantastic. So I just wanted to say it's not all darkness, not all bad." Other participants used non-broken hearts to represent self-love and hope. Kenzie (cisgender woman, White, USA), drew hearts to embody self-appreciation and to acknowledge that she is a "loving person who cares for people." While positive imagery was less common,

it is important to highlight given the otherwise overwhelming negative depictions in the body maps of life with HS.

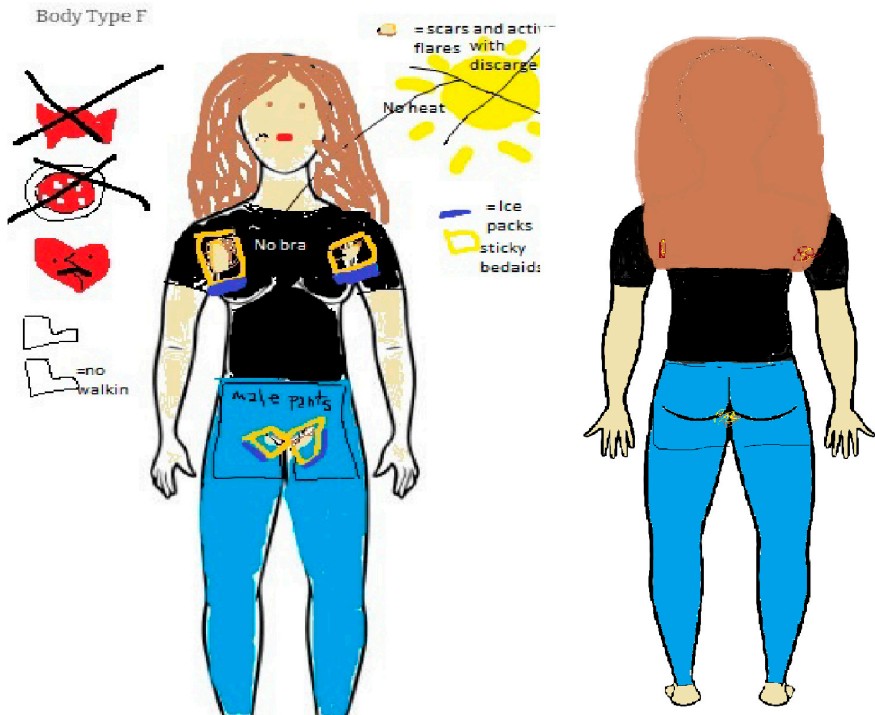

**Figure 9.** Laura's body map.

Finally, participants sometimes drew literal objects associated with their HS experience. Sadie (cisgender woman, White, USA; Figure 10) drew bandages to represent the need for wound care with HS flares, a scalpel to describe the "worst pain of [her] life" from surgical procedures on her HS flares, and a maroon and gray injection pen for her biologic medication. She also used symbolic images such as hands on her groin that represented a "protectiveness" over that area of her body. Laura (Figure 9) also used a blend of literal and figurative elements in her body map, drawing actual foods she has to avoid due to her HS (candy and pizza), boots to show limited walking ability, and ice packs and "sticky bandaids" under her arms and in her groin to help control the pain and drainage from HS flares. Laura also used some figurative imagery, like the sun to represent avoiding excessive heat. Participants described HS-related objects in their in-depth interviews, and these objects' appearance in the body maps reflects important and salient aspects of their HS lived experience.

*3.4. Words on Body Maps*

Although our conceptualization of the body map was that the participants would use color and images to illustrate their experience of living with HS, twenty-six participants (86%) used words in combination with drawings to describe their HS experience. At least half of the participants provided illustrative or text-based additions to the body maps that described living with HS in more detailed, rich terms than the interview conversation alone. We did not find any relationship between color and written words; most were written in black for readability and easy access to black pens or markers over other colors.

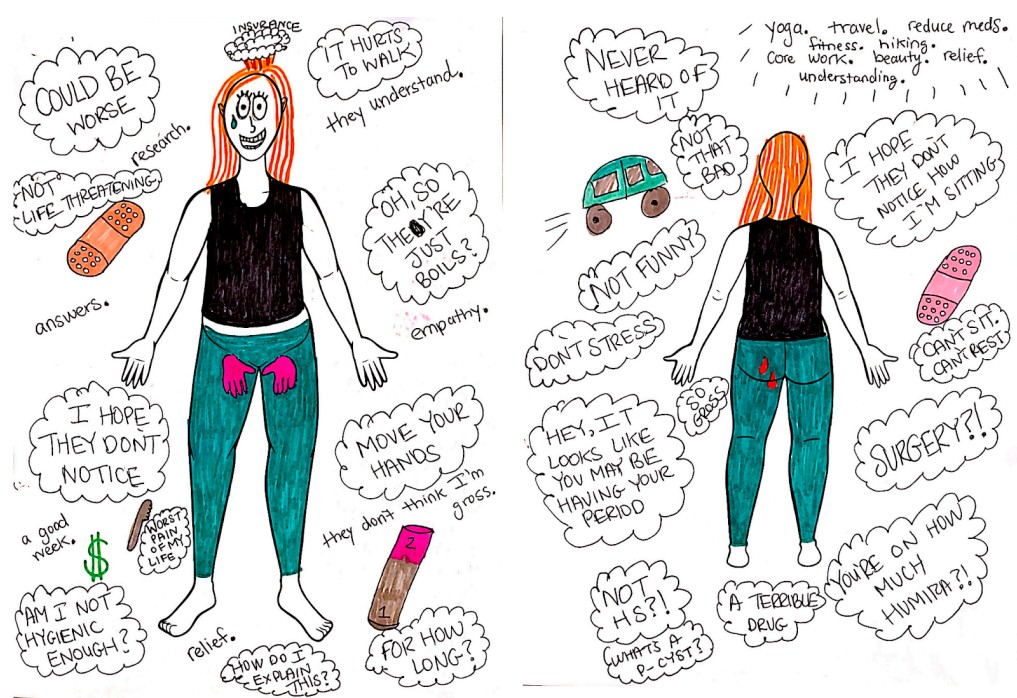

**Figure 10.** Sadie's body map.

Words were often written around the body outline shape but were sometimes written on parts of the body where a specific emotion was attached. For example, Sadie used word bubbles surrounding her whole body map in addition to images (Figure 10). She explained this approach by saying, "I felt like the best way to show some of the other parts of my HS are just through words...some of these things are like statements that people have said to me. Some of them are like thoughts that I've had. And then the stuff that's like outside of the bubble is like more positive things, I guess." Here we see examples of what Charmaz (1983) calls "discrediting definitions of self" that arise in interaction with others when chronically ill people experience public embarrassment or shame because of their condition. Sadie describes how people commented on her bleeding wounds, "Hey, it looks like you may be having your period" or had negative commentary about her Humira medication. Charmaz illuminates how the discrediting can be even more significant with close confidants or if comments occur more frequently. Sadie's close friends in college dismissed her symptoms as "not very serious" which made her doubt the difficulties she faced as a result of her HS. Sadie struggled to get started with her body map and said that writing words and phrases helped her process her thoughts and decide what images to use.

Participants who do not consider themselves to be "artsy" may have chosen to write words, create simple drawings, or use plain colors for convenience reasons. This led to several participants writing only a few words or no words at all, but rather just indicating the location of their HS flares with circles or dots.

*3.5. Body Maps as Medical Drawings*

Half of the participants used the body map as a medical illustration. Participants included circles or dots, sometimes in red but often in black or blue pen, to indicate the areas impacted by HS. For most of them, this was their underarm/axillae, groin (labia, mons, or genital region more generally), inner thighs, or breast area. Sarah (Figure 4) had what we coded as a medical-style body map, only indicating dots where her HS was located, though she did add several colors to represent various aspects of her HS experience. Anastasia (cisgender woman, Latina/Hispanic, USA) described technology challenges that led to her more simplistic, medical illustration style of body map (Figure 11). She tried to access the map files at work but was blocked from downloading them and she did not have

a printer at home. She also noted the limitation of not having the body outline with lifted arms to be able to show HS flares in the armpits, a key area of HS symptoms for her.

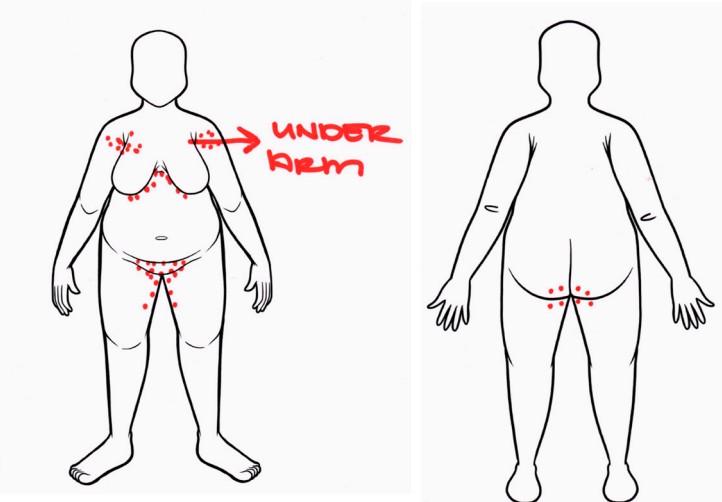

**Figure 11.** Anastasia's body map.

One participant assumed that medical providers were working on this study, even though we did not indicate this in the informed consent or study website, so he thought the body map was going to be used to determine the severity of his condition. This highlights another limitation of communicating electronically about a visual form of data collection; several participants made assumptions about the purpose of the study that were outside of what our materials described. We did not include an in-depth analysis of how participants with medical drawing body maps illustrated their body maps because they did not meet the original intent of the body mapping method. However, we did feel that it was important to include them in the results to give context to why they may have chosen this style of illustration.

*3.6. Participant Reflections on Body Mapping Process*

The researchers asked participants about their body mapping experiences during in-depth interviews. Overall, participants responded positively to the body mapping process and many shared how it helped them make sense of their HS in new ways, which helped both personally and in communicating about their experience of the disease with others. Crystal (cisgender woman, no race identified, USA) described her plans to share her body map with her wife to help deepen her wife's understanding of her lived experience:

*I'm gonna share this with my wife and show, you know, show her as well. Even though she understands and sees [the HS] I feel like she doesn't always understand, so visually I think this would help her get it, you know, so yeah…. It's been, it's been a very good learning and therapeutic experience.*

The creative aspect of body mapping intimidated some participants at first, but they were able to tap into artistic expression in a way that stayed with them even after completing the body map. Ernesto (Figure 2) described starting the body mapping process several times and throwing away his first few attempts before settling into his own approach.

*And what really helped me was trying to get the story back. To really try to find, uh, I think storytelling really works for me, you know, so uh like trying to see what has been happening, and how can I tell this story with the body map. So I finally got something for me and just kept everyone away, and just took my colors and starting. And yeah, then it was easy. No, but I think many ideas have been working inside of me for a couple days.*

Other participants shared how body mapping gave them room to think deeply about their HS experiences over time in a way they never had before. Eliza (Figure 7) thoughtfully

described how the slower, introspective nature of body mapping translated her HS into a tangible journey through physical and emotional experiences.

> *I don't know that I've ever thought about the process… I live the process, I'm busy. I just, you know, I just plug along, plug along, plug along. So taking a moment just to sit and think about it. I think it's probably the first time that I've said, like you know what actually I've come a long way with this. Uhm. And it's so easy if I talk about it, if I bring it up or if I'm you know, engaging with somebody about it for first time. Because the initial emotions I felt when… I had my first flare are still so visceral to me. That's easily what I described, right. I don't describe where I am now, I describe the first flare. The confusion, the fear. So it was nice to like to honor that. And say that's a part of my journey and is not, should not be minimized. But I'm also in a different place now and I don't know that I've ever taken the time to see the journey.*

Twenty of the fifty interview participants declined to participate in body mapping, with several citing that they either did not feel "artistic enough" to express their experiences of HS or technically literate enough to navigate the silhouette download process. Others just did not want to engage in the process. Of those who did illustrate body maps, several cited barriers to fully engaging with the artistic dimension of the body mapping method. For example, Laura (cisgender woman, White, USA) described the creative limitation and how it led her to focus on using words and red dots rather than more complicated images on her body map. She noted that this was related to a lack of artistic ability and that choosing the body size of the silhouette was difficult because none were perfectly aligned with her body. Her overall experience of body mapping presented an emotional challenge that she was able to overcome through completing the task: "I'm not going to lie it was, um, it was a little emotional. But it wasn't devastating or catastrophically so, but it was just, but it felt good once I got it done and sent off to you too." Participant resistance to complete body maps may also reflect our use of written instructions via email, while traditional body mapping involves in-person, interactive methods with materials supplied directly to participants (Boydell 2021).

In contrast, Sarah (Figure 4) highlighted how she overcame her initial hesitation with the body map and ended up enjoying the process.

> *I was like, it was a little strange to put it on paper cause like for once someone was actually asking me about it and that was kind of weird because no one ever asks. No one really wants to talk about abscesses on your skin. You know it kind of makes you sound like a leper, you know. I thought it was kind of empowering to fill this out. I was excited.*

For those who did complete the body maps, many expressed how the process allowed them to critically think about their HS illness in ways they had not considered previously. While the maps were intimidating to some, and others chose not to complete them at all, the simple act of being asked about their HS experiences yielded positive responses for most of the participants.

## 4. Discussion

Body maps produced by participants yielded robust data about their embodied experiences with HS, highlighted how different people interpret and create visual data about the disease, and showed how the body mapping approach could elucidate HS experiences in a new way. Unlike previous case studies of visual methodologies in health and illness literature (Phillips et al. 2015; Guillemin 2004), we did not have a blank slate starting point for our participants' drawings of their lived experience of HS. Our study drew from body mapping research (Gastaldo et al. 2018; De Jager et al. 2016) to adapt our approach and provide a set of body silhouettes for participants to choose from rather than tracing their own bodies on large pieces of paper, which would not have been conducive to our international, virtual data collection study design. This paper's analysis of the actual body maps, not just the process of producing the body maps, responds to calls for more results and themes from analyzed body maps in the literature (Coetzee et al. 2019; De Jager et al.

2016). To our knowledge, this is the first body map study to use pre-produced silhouettes reflecting different body sizes and shapes. We are also the first study to use body maps to explore HS experiences.

Visual data from the body maps included powerful colors, images, and words to describe the embodied experience of people living with HS. While many participants approached the body map illustrations with the intended artistic approach outlined in the research instructions, some used their maps as medical drawings to indicate areas impacted by HS. Several participants illustrated both their physical symptoms and drew symbols, words, and even poems to capture the nuance of their HS experience beyond the physical body. HS can be both physically and emotionally painful, and visual representations to describe pain align with themes from other body map studies about injuries, chronic illness, and chronic pain (Van Schelven et al. 2023; Santarossa et al. 2023; Ludlow 2014; Tarr and Thomas 2011). Participants described the deep emotional impacts of HS, including feelings of sadness, isolation, frustration, shame, and missing out on important aspects of life. Some also highlighted areas of hope and gratitude. These reflect themes in the existing social science literature about HS's influence on lived experiences and quality of life (Howells et al. 2021; Ingraham et al. 2022).

The body map illustrations of lived experience with HS also contribute to the literature on illness narratives, particularly how chronic illness can serve as a "biographical disruption" (Bury 1982) and impact one's sense of self, especially around loss of self in social interaction (Charmaz 1983, 1991). The restrictions that HS places on the lives of participants, represented visually by jail bars, chains, thorns, or cross-out symbols of major life goals, demonstrate the loss they feel of the life they either had previously or could or should have without this chronic illness. The couple of participants who represented the passage of time related to their HS diagnosis demonstrated the continued importance of Bury's (1982) notion of biographical disruption. However, this is often more complicated for HS patients, as many see the onset of the illness as the disruption of their life trajectory rather than the diagnosis itself, given that HS can sometimes take years to diagnose (Aparício Martins et al. 2023). This study also builds on previous applications of biographical disruption in dermatological conditions (Jobling 1988; Al-Muhandis 2024) that importantly highlight how disfigurement is somewhat unique to these conditions. Future research on dermatological conditions should also take up Jobling's (1988) focus on the connections between the biographical disruption, disfigurement, or visibility of these conditions and its connection to stigma, which was beyond the scope of this article.

We found that body mapping as a component of qualitative in-depth interviews allowed participants to express parts of their lived experience with HS in ways that verbal prompts did not. Participants reported that they enjoyed it as both a creative activity and as a method of processing how they felt about their HS across the lifespan of their condition. Body mapping facilitated a unique understanding of their own HS journeys while providing new ways of communicating about this chronic illness with others, including loved ones. Interviews revealed that body maps could be a useful tool in the mental health support of HS patients given their increased risk of depression and suicidality (Patel et al. 2020). This reflects research that recommends body mapping in art therapy and other studies that found that body mapping had a therapeutic effect even if therapy was not the intended goal of the study (Santarossa et al. 2023; Mayra et al. 2022). However, body mapping as potential mental health support was also a limitation in our study. Our research interviews—where body maps were discussed—were not designed to be therapeutic, and the interviewers did not have the capacity to help participants process strong emotions that may have come up while creating the body maps. The research team provided a list of resources available to people living with HS on the study website and to participants individually in the consent materials. Future application of body maps in HS research, and for any chronic condition, should plan for this potential therapeutic effect and include checkpoints along the way to connect participants with appropriate mental health resources.

The adapted body mapping method used in this study generated confusion for some lower technology literate participants who did not understand how to download the body map files or edit them to produce the drawings, despite our written instructions and follow up communication offering them assistance. This is an important consideration for older, low literacy, and low technology access populations impacted by HS whose perspectives are not represented here. The study is also limited by the convenience, community-based sample (rather than a clinical sample) with an overrepresentation of White, cisgender women from the United States despite attempts at purposive oversampling of women of color and cisgender men. However, the 31 completed body maps with accompanying interviews reflect a diversity of body sizes and shapes, which is new for both HS and body map research.

Future research at the intersection of dermatological conditions and body mapping is certainly warranted, as skin conditions are a unique type of chronic illness that are visible in ways that many other illnesses are not. Using the pre-existing body silhouettes was helpful for most participants who did not feel artistic enough to render their own body shape but limiting for participants who did not find a body shape that reflected them. The body silhouettes were also limiting for specific HS experiences because they did not illustrate underarm areas with raised arms, which was a key site of HS for almost all participants. It was also difficult for participants to illustrate HS flares on their inner thighs, groin, or genitals with the pre-drawn silhouettes. Illustrating flares was not the original goal of this research, but it was important for participants as part of their body mapping process. We would suggest adapting the silhouettes to include one raised arm and including more language to reassure participants that artistic ability is not required for the body maps, a common theme in other body map studies (Henderson et al. 2023; de Souza et al. 2021). Because this was the first body map study for people with HS, future studies should reiterate the purpose of body mapping, remind participants that it is more than just indicating where HS symptoms appear on the body, and consider video recording the instructions so participants can follow along with the prompts more easily. Researchers could also consider the timing of when the body map is illustrated (Coetzee et al. 2019). For example, a post-interview approach suggested by Brailas (2020) has participants draw the body map in real time during the interview and describe and discuss it as part of the main interview process.

Outside of research, body mapping as an emotional process tool could be useful for provider-facilitated or community-based advocacy support groups that meet virtually or in person. This would restore body mapping to more of its art therapy roots and connect participants with the mental health benefits of body mapping (Murray et al. 2023). These tools could be used in HS support groups to help process the emotional toll of HS and help providers understand patients' mental health or pain management needs more holistically.

## 5. Conclusions

This study utilized body mapping methodology to understand the lived experiences of 31 people with hidradenitis suppurativa (HS) within a mixed method research project. The body maps helped participants understand and express their HS experiences in unique ways that cannot always be expressed verbally or in writing. This study adds to the limited social science literature about HS and introduces body mapping as a relevant person-centered qualitative method for exploring chronic dermatological conditions.

**Author Contributions:** Conceptualization, N.I. and L.R.H.; Methodology, N.I. and L.R.H.; Formal Analysis, N.I. and K.D.; Investigation, N.I. and L.R.H.; Resources, N.I. and L.R.H.; Data Curation, N.I., L.R.H. and K.D.; Writing—Original Draft Preparation, N.I., K.D. and L.R.H.; Writing—Review & Editing, N.I. and L.R.H.; Visualization, N.I. and K.D.; Project Administration, N.I. and L.R.H.; Funding Acquisition, N.I. and L.R.H. All authors have read and agreed to the published version of the manuscript.

**Funding:** The Augustana College Faculty Research Grant awarded to Lena Hann helped fund a portion of the data collection and analysis. The California State University Faculty Research grant awarded to Natalie Ingraham funded participant incentives and research assistant salary for 2022.

**Institutional Review Board Statement:** The study was conducted in accordance with the Declaration of Helsinki, and approved by the Institutional Review Board of California State University East Bay (protocol code CSUEB-IRB-2020-108 on 29 October 2020).

**Informed Consent Statement:** Informed consent was obtained from all subjects involved in the study using verbal consent before the interviews began. Written informed consent was obtained from the participants for body map publication via email.

**Data Availability Statement:** The datasets presented in this article are not readily available because the data are part of an ongoing study. Requests to access the datasets should be directed to the corresponding author.

**Acknowledgments:** The authors would like to thank the research assistants who helped with data collection and cleaning: Caleb Drew, Abraham Jalala, and Kody Walker. Jamie Harris also assisted with data cleaning and interview transcript analysis.

**Conflicts of Interest:** The authors declare no conflicts of interest.

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
