# Peer review of "“It’s Like Having an Uncontrolled Situation”: Using Body Maps to Understand the Embodied Experiences of People with Hidradenitis Suppurativa, a Chronic Dermatological Condition"

_socsci, doi:10.3390/socsci13030168_

Round 1
Reviewer 1 Report
Comments and Suggestions for Authors
It was a pleasure to have the opportunity to read this manuscript, reporting on the use of body mapping to explore and depict people's experiences of Hidradenitis suppurativa (HS), of which I previously knew nothing. I firstly thought it commendable that the authors decided to pursue the choice to use body mapping, considering the conditions of working during the pandemic, and having to alter this method to an online research process. It was incredible how participants still took to this method and used the resources they had available to portray their experiences. It also emphasised how eager they must have been to get an opportunity to talk about their experiences beyond the limited setting of an online community of peers with the same condition.
The contextual and methodological background was very clearly defined and explained, citing key literature in the field. The application of the method within the conditions of the pandemic was highly commendable and offers opportunities for future applications with other similar populations.
I was unclear on what the overarching theoretical lens of the project was, in terms of what guided the analysis - particularly with the mixture of deductive and inductive coding - so would be interested to know what theory was applied to the analysis of the data.
The study was reported as 'mixed-methods' but I did not get any sense of what component made up the quantitative analysis or how this interacted or combined with the qualitative analysis. Was this part of a larger study, with quant results reported elsewhere? Could the authors please explain?
I also was not sure why the authors chose to use percentages (e.g. stating '43% of participants...'), instead of just citing number of participants, to refer to the quantity of participants who reported an experience or did something within their interview (e.g. using blue to depict something). I thought this was an odd choice within a primarily qualitative study, and would appreciate some insight into this or suggest the authors revert to quoting actual numbers of participants.
Please remove the lines 241-243 which must have been left in from the publishing template.
It was noted that the research interviews were limited in terms of their capacity for therapeutic value, which is entirely understandable. It would be useful to know, however, how participants responded to participating in the study and whether any indicated a need to be referred to formal support. Please could the authors indicate this?
It was noted that the pre-determined body silhouettes limited how participants could depict or explore their illness flares. Would the authors still choose to pursue using pre-determined body maps in future, over hand-drawn ones? This would be useful to know for other researchers in related studies/issues/contexts.
Apart from these minor points, I believe this manuscript offers considerable value for providing insight into the lived experiences of people who live with a highly stigmatising chronic condition, using a method which clearly allows us insight into such experiences.
Reviewer 2 Report
Comments and Suggestions for Authors
